# Genetic Profiling of Cell-Free DNA in Liquid Biopsies: A Complementary Tool for the Diagnosis of B-Cell Lymphomas and the Surveillance of Measurable Residual Disease

**DOI:** 10.3390/cancers15164022

**Published:** 2023-08-08

**Authors:** Gloria Figaredo, Alejandro Martín-Muñoz, Santiago Barrio, Laura Parrilla, Yolanda Campos-Martín, María Poza, Laura Rufián, Patrocinio Algara, Marina De La Torre, Ana Jiménez Ubieto, Joaquín Martínez-López, Luis-Felipe Casado, Manuela Mollejo

**Affiliations:** 1Department of Haematology, Hospital Universitario de Toledo, Av. del Río Guadiana, 45007 Toledo, Spain; lparrilla@sescam.jccm.es (L.P.); marinadelatorredelapaz@sescam.jccm.es (M.D.L.T.); fcasadom@sescam.jccm.es (L.-F.C.); 2Altum Sequencing SL, Av. Gregorio Peces Barba, 1, 28919 Madrid, Spain; alejandro_martin@altumsequencing.com (A.M.-M.); santiago_barrio@altumsequencing.com (S.B.); laura_rufian@altumsequencing.com (L.R.); 3Computational Science Department, Carlos III University, Ronda de Toledo, 1, 28005 Madrid, Spain; 4Biobank Department, Hospital Universitario de Toledo, Av. del Río Guadiana, 45007 Toledo, Spain; ycampos@sescam.jccm.es; 5Haematology Department, Hospital Universitario 12 de Octubre, Avda. de Córdoba, s/n, 28041 Madrid, Spain; mpoza@salud.madrid.org (M.P.); ana.jimenezub@salud.madrid.org (A.J.U.); jmarti01@med.ucm.es (J.M.-L.); 6Genetics Department, Hospital Universitario de Toledo, Av. del Río Guadiana, 45007 Toledo, Spain; paplana@sescam.jccm.es; 7Anatomopathology Department, Hospital Universitario de Toledo, Av. del Río Guadiana, 45007 Toledo, Spain; mmollejov@sescam.jccm.es

**Keywords:** B-cell lymphoma, liquid biopsies, cfDNA, NGS, minimal residual disease, PET/CT

## Abstract

**Simple Summary:**

Genetic profiling of plasma cell-free DNA (cfDNA) from liquid biopsies (LiqBio) is a possible alternative to genetic profiling of tissue biopsies to diagnose B-cell lymphoma (BCL), and a complementary tool of PET/CT to determine measurable residual disease (MRD). The aim of our study was to confirm the utility of LiqBio for diagnosis and surveillance of BCL. We confirmed the correlation between somatic mutations in paired LiqBio and tissue biopsies at diagnosis in a population of 78 patients; furthermore, we identified additional mutations in LiqBio at diagnosis from patients with no available tissue samples or no mutations detected in tissue samples. As a surveillance tool for MRD detection, LiqBio combined with PET/CT showed to be a valuable method, improving the PET/CT specificity.

**Abstract:**

Purpose: To assess the potential value of LiqBio as a complementary tool for diagnosis and surveillance of BCL. Methods: This prospective multi-center study included 78 patients (25 follicular lymphomas (FL) and 53 large B-cell lymphomas (LBCL)). We performed next-generation sequencing (NGS) of cfDNA LiqBio and paired gDNA tissue biopsies at diagnosis and compared the mutational statuses. Also, through NGS of LiqBio, we identified MRD biomarkers and compared this novel LiqBio–MRD method with PET/CT in detecting MRD at follow-up. Results: We identified mutations in 71% of LiqBio and 95% of tissue biopsies, and found a correlation between variant allele frequency of somatic mutations. Additionally, we identified mutations in 73% of LiqBio from patients with no available tissue samples or no mutations in them. Regarding the utility of LiqBio–MRD as a dynamic monitoring tool, when compared with the PET/CT method, a lower sensitivity was observed for LiqBio–MRD at 92.3% (vs. 100% for PET/CT), but a higher specificity of 91.3% (vs. 86.9% for PET/CT). Conclusion: Genetic profiling of tumor cfDNA in plasma LiqBio is a complementary tool for BCL diagnosis and MRD surveillance.

## 1. Introduction

Mature B-cell neoplasms comprise more than 80% of non-Hodgkin lymphomas and consist of different pathological subtypes with variable clinical outcomes [1]. Tumor tissue biopsies are the gold standard to analyze lymphoma genetic profiles, identifying useful mutations as molecular biomarkers [2,3]. Moreover, targeted next-generation sequencing (NGS) has contributed to reclassifying molecular subtypes of lymphomas through new algorithmic models. These subtypes differ in clinical outcomes and in responses to therapies targeting oncogenic signaling pathways and immunotherapies, allowing better knowledge and management of the disease [4,5,6,7,8]. However, biopsies are invasive, costly, and might not reflect the entire molecular complexity of these heterogeneous tumors. Recently, genetic profiling of plasma cell-free DNA (cfDNA) from liquid biopsies (LiqBio), containing circulating tumor DNA, has emerged as a promising minimal invasive alternative to genetic profiling of tissue biopsies [3,9,10,11,12,13,14,15,16,17]. Currently, this tool is being assessed in the diagnosis, prognosis, response evaluation, and follow-up of B-cell lymphomas (BCL) [9,18,19,20,21,22,23], including primary cerebral nervous system lymphoma (PCNSL) [23,24,25]. Due to the limitations of computed tomography (CT) and positron emission computed tomography (PET/CT) in determining measurable residual disease (MRD), recent studies have investigated the combination of the analysis of cfDNA from LiqBio with PET/CT to monitor MRD and predict relapse in patients with BCL [19,26].

In this study, we checked the hypothesis that genetic profiling of plasma cfDNA from LiqBio was a complementary tool to genetic profiling of tissue biopsies to diagnose BCL, and to PET/CT to detect MRD. To this aim, we used targeted NGS to compare the mutational status of cfDNA from plasma samples with the mutational status of gDNA from tissue samples of patients with large BCL (LBCL) or follicular lymphoma (FL) at diagnosis. Also, through NGS of LiqBio, we identified MRD biomarkers and evaluated the utility of this novel LiqBio–MRD method to detect MRD, in comparison to PET/CT.

## 2. Materials and Methods

### 2.1. Patient Cohort and Sample Collection

This was a prospective, observational, non-interventional, multi-center study. The cohort included 78 newly diagnosed or recurrent patients with LBCL or FL. The study was conducted at the Hospital Universitario de Toledo (Toledo, Spain) or Hospital Universitario 12 de Octubre (Madrid, Spain), between January 2020 and September 2022, except for three patients recruited in 2017 as experimental cases. Lymphomas were classified according to the 2017 World Health Organization (WHO) classification [27]. We included formalin-fixed paraffin-embedded (FFPE) tissue samples at the time of diagnosis (*n* = 59) and pre-treatment plasma samples (*n* = 70): 51 patients presented paired plasma cfDNA and tissue samples at baseline, 19 patients only presented plasma cfDNA samples, and 8 patients only tissue samples. Subsequently, we collected plasma samples during the follow-up (*n* = 47) at different time-points: after two cycles of therapy, at the end of therapy (EOT), and after therapy as follow-up, or in case of progression or relapse (Figure 1). A total of 264 samples were collected. We selected somatic mutations of samples at diagnosis as biomarkers for disease monitoring by LiqBio–MRD during the follow-up. We evaluated responses by PET/CT or CT together with clinical assessment.

The study was approved by the Ethics Committee of the Hospital Universitario de Toledo (Spain), and informed written consent of all patients was obtained according to the Declaration of Helsinki.

### 2.2. DNA Purification and Quantification

Solid tissue DNA was extracted with a QIAamp gDNA FFPE kit (Qiagen, Hilden, Germany) using two to four sections from 5 to 10 µm, cut from the original paraffin block. Then, the gDNA was eluted in 35 μL ATE buffer and quantified using the Qubit BR kit (Thermo Fisher Scientific, Waltham, MA, USA). For cfDNA extraction, 10 mL of peripheral blood were collected in Streck or Roche Cell-Free DNA collection tubes (Roche Diagnostic, Basel, Switzerland) for plasma separation and cfDNA purification. The plasma was separated with two centrifugation steps at 1600× *g* and 4500× *g*, stored at −80 °C, and subsequently sent to the Hospital Universitario 12 de Octubre (Madrid, Spain). The purification of cfDNA was performed with a QIAamp Circulating Nucleic Acid kit (Qiagen) and quantified using a Qubit HS kit (Thermo Fisher Scientific). On average, 17 ng of cfDNA/mL of plasma (rank 1.2–852.2 ng/mL) was obtained from the initial 10 mL of peripheral blood. Fragment size and genomic DNA contamination were quantified using a Bioanalyzer 2100 fragment analysis system (Agilent, Santa Clara, CA, USA).

### 2.3. Baseline Genotyping and LiqBio–MRD Biomarker Selection

The solid tissue gDNA and plasma cfDNA baseline samples were screened for mutations with a short-length AmpliSeq Custom Panel (Thermo Fisher Scientific). The panel was already established as a routine diagnosis tool at the Hospital Universitario 12 de Octubre (Madrid, Spain) and was designed to cover all coding regions of 56 lymphoma-specific genes in the FFPE samples (Appendix A). Samples were sequenced with an average coverage of 2150× on an Ion S5 System platform (Life Technologies, Thermo Fisher Scientific). Variant annotation was performed using the default annotated variants single sample workflow from the Ion Reporter software (version 5.18.2.0). In the case of FFPE samples, deamination-related base changes were reduced by filtering out C > T/G > A changes with a frequency below 20% and a transformed *p*-value greater than −2, unless previously described as a somatic aberration in lymphoma (COSMIC database). Only somatic mutations previously described in lymphoma or similar cancers were used as MRD biomarkers.

### 2.4. LiqBio–MRD Methodology and Bioinformatic Pipeline

A multiplexed mini panel was defined for every patient with the specific MRD biomarkers identified at diagnosis. The mini panels included molecular-tagged primer pairs to amplify every mutation in three biological replicates. Each primer generated three types of reads. The targeted sequencing depth was 500,000× per amplicon. The MRD libraries and sequencing were conducted as previously described [28]. The FASTQ files were automatically analyzed and demultiplexed via a customized bioinformatic pipeline, programmed in Python and R. To compute the MRD ratio for each genetic position, the aligned wild-type and mutated sequences (with a margin of 15 bp, queried to Ensembl through its Python API) were obtained from the corresponding demultiplexed output file. First, the MRD ratio was generated independently for each triplicate. Then, noise effects arising from PCR and sequencing were controlled by identifying and removing artifacts that were above the mean ratio plus one standard deviation (SD). Finally, the corrected mean MRD ratio was compared with the limit of detection (LOD) and limit of quantification (LOQ) calculated for each mutation independently in three triplicates of three healthy donors. The LOD was computed as the mean ratio in control samples plus three times the SD and, for LOQ, plus ten times the SD. Every hotspot with a corrected mean MRD ratio below the LOD was automatically eliminated. The final LiqBio–MRD value was defined by the mutation with the highest MRD ratio at the sampling time-point. The sensitivity of the test was 2 × 10^−4^ as previously defined [19].

### 2.5. Statistical Analysis

The Pearson correlation coefficient was used to determine the linear relationship between the allele frequency of each mutation detected in both fractions (FFPE and plasma). The Mann–Whitney U test was used to determine statistically significant differences in the LiqBio–MRD values between the PET/CT-negative and -positive categories. A chi-square test of independence of variables in a contingency table applying Yates’ correction was performed between cfDNA concentration categories (higher or lower than 5 ng of cfDNA per mL of plasma) and different clinical categories. All these tests were performed using Python (version 3.10.8), with the Python package SciPy (version 1.10.1); *p*-values of ≤0.05 were considered statistically significant.

## 3. Results

### 3.1. Patient Characteristics

The characteristics of 78 patients with BCL are shown in Table 1. The median age was 62 years (range 19–89 years) and 54% of patients (42) were females. In total, 25 were diagnosed with FL and 53 with LBCL. Of the latter, 27 presented diffuse LBCL, not otherwise specified (DLBCL, NOS); 12 were primary mediastinal large BCL (PMBCL); 7 were PCNSL; 5 were high-grade BCL with MYC and BCL2 and/or BCL6 rearrangement; 1 was EBV-positive diffuse large BCL, NOS; and 1 was BCL, unclassifiable, with features intermediate between DLBCL and classic Hodgkin lymphoma (gray zone B-cell lymphoma). More than half of the patients (44) had an advanced stage III-IV at diagnosis.

Most of the patients, except for six (CHT028, CHT034, CHT051, CHT059, CHT065, CHT068), were treatment-naïve at their enrollment. A percentage of 4% (3/78) of patients were not treated: one was under a watch and wait approach, and two were unfit for treatment. Seventy five patients (96%) received treatment with chemotherapy: thirty-four with rituximab–cyclophosphamide, doxorubicin, vincristine, and prednisone (R-CHOP); eighteen with etoposide, prednisone, vincristine, cyclophosphamide, and rituximab (DA-EPOCH-R); ten with rituximab and bendamustine (R-Benda), one with rituximab, cyclophosphamide, vincristine, and prednisone (R-COP); seven with a high-dose methotrexate-based regimen for PCNSL; one with rituximab, etoposide, methylprednisolone, high-dose cytarabine, and cisplatin (R-ESHAP); one with polatuzumab vedotin, bendamustine, and rituximab (pola-BR); one with rituximab in monotherapy (R); and two with local radiotherapy as monotherapy (RT). Thirteen patients received consolidative local radiotherapy. The median follow-up was 19 months (range 0–69). Out of the 75 patients who underwent treatment, 51 (68%) achieved complete response, 21 (28%) developed a partial response or progressive disease, and three (4%) were lost to follow-up or deceased because of infectious complications during therapy.

### 3.2. Baseline Genotyping of Plasma Samples (cfDNA) and Tissue Samples (gDNA)

We analyzed gDNA from 59 tissue samples and cfDNA from 70 plasma samples at the time of diagnosis using a targeted NGS panel (Figure 2). With this baseline of tissue genotyping combined with plasma samples, we could detect baseline mutations in 92% of patients (72/78). Specifically, the mutation rate was 95% (56/59) in the gDNA samples and 71% (50/70) in the cfDNA samples. In the plasma samples, detection differed among lymphoma subtypes: 82% (27/33) in DLBCL, 86% (6/7) in PMBCL, 43% (3/7) in PCNSL, and 56% (14/25) in FL. The mean of cfDNA concentration was 17 ng/mL (range 1.2–852.2): 20.7 ng/mL in LBCL, 17.65 ng/mL in PCNSL, and 10.55 ng/mL in FL. The mean of variant allele frequency (VAF) was 0.26 (0.03–0.98) in tissue samples, and 0.18 (0.03–0.88) in the plasma samples. In total, we could identify 437 somatic mutations in the tissue samples, with a median of 7.41 mutations (range 0–28) per sample; and 209 somatic mutations in the plasma samples, with a median of 2.99 mutations (range 0–14) per sample. We did not detect any mutations in 8% of the tissue samples (5/59 three were FL and two LBCL), and in 29% (20/70) of the plasma samples. However, within the cases with no mutations detected in the tissue samples (4) or with no available paired tissue samples (18), we were able to identify suitable alterations for MRD monitoring in 73% (16/22) of the plasma samples. Among the 51 patients with paired tissue and plasma samples at diagnosis, we identified 379 mutations: 115 were identified in both samples, 230 were only detected in tissue samples, and 34 only in plasma samples.

To evaluate genomic variations between LBCL and FL, we compared somatic mutations identified in different pathological subtypes (Figure 2). The most recurrently mutated genes in tissue and blood samples of LBCL were the following: *KMT2D* (32%, 17/53)*; TP53* (30%, 16/53); *MYD88* (22%, 12/53)*; CARD11* (21%, 11/53)*; CREBBP* (15%, 8/53)*;* and *BCL2* (15%, 8/53). In particular, *CARD11* and *KMT2D* predominated in the germinal center (GC) phenotype, and *MYD88*, *PIM1*, and *CD79B* in non-GC phenotype. In PMBCL, the most frequently mutated genes were *STAT1* (83%, 10/12) and *SOCS1* (66%, 8/12); and in PCNSL, *MYD88* (80%, 4/5). On the contrary, in FL, the most frequently mutated genes were *KMT2D* (64%, 16/25), *CREBBP* (36%, 9/25), *BCL2* (32%, 8/25), and *EZH2* (24%, 6/25). We observed a higher correlation of the VAF of somatic mutations identified in the paired blood and tissue samples in LBCL (*r* = 0.375, *p* = 0.001) than in FL (*r* = 0.459, *p* = 0.032).

### 3.3. Association between cfDNA Baseline Levels and Clinical Features

We first evaluated the correlation between the amount of cfDNA at the baseline with various clinical parameters. A correlation was identified between cfDNA levels and bulky disease, lactate dehydrogenase (LDH) levels, and LBCL subtype (*p* < 0.05). This suggested that higher levels of cfDNA (>5 ng/mL) were significantly associated with a larger tumor size or more aggressive subtypes. No significant correlations were found between cfDNA levels and Ann Arbor stage, extranodal extension, ECOG scale, β2-microglobuline, hemoglobin levels, or bone marrow infiltration (*p* > 0.05).

### 3.4. Correlation between PET/CT and LiqBio–MRD

We then evaluated the correlation between the results of PET/CT and of LiqBio–MRD. Thirty-six out of seventy-eight patients had both simultaneous PET/CT and cfDNA samples available: thirty-one at EOT, and five at follow-up. On the basis of PET/CT results, we defined two groups: (1) a negative group including patients with complete response (Deauville score 1–3, 20/36); and (2) a positive group including patients with partial metabolic responses (Deauville score 4–5 with reduced FDG intensity in comparison with baseline, 5/36) and patients with progressive disease (Deauville score 5 with an increase of FDG intensity, 11/36) (Figure 3). There was an 83% (30/36) concordance between PET/CT and LiqBio–MRD (18 patients were negative and 12 were positive according to both techniques), and six patients presented discordant results (Figure 3). Within the six discordant cases, four patients had a positive PET/CT and negative LiqBio–MRD. Of these four patients, three had a negative PET/CT after six months, confirming complete response. However, one out of the four patients transformed Hodgkin lymphoma after four months from the EOT. The two discordant cases with negative PET/CT and positive LiqBio–MRD were in complete response after 15 months of follow-up. These cases had two characteristics in common: (1) Their LiqBio–MRD was positive because of a mutation in *TP53*, and (2) both had a history of an epithelial carcinoma before or after the lymphoma was diagnosed. Therefore, of the six cases with discordant results (17%, 6/36), three were false positives by PET/CT, two were false positives by LiqBio–MRD, and one was a false negative by LiqBio–MRD. Finally, we observed a lower sensitivity for LiqBio–MRD (92.3% vs. 100% for PET/CT), but a higher specificity (91.3% vs. 86.9% PET/CT).

### 3.5. Dynamics during Follow-Up cfDNA (LiqBio–MRD): Representative Cases

Four representative cases of different scenarios are shown in Figure 4. An example of disease remission with concordant results of PET/CT and LiqBio–MRD can be observed in patients CHT028 (Figure 4A) and CHT049 (Figure 4B), with subsequent relapse in patient CHT028. In patient CHT005 (Figure 4C) no PET/CT interim was available because of the COVID-19 pandemic emergency. After a negative post-2-cycle-therapy LiqBio–MRD, a third sample at EOT tested positive, predicting the relapse that occurred three weeks later. In patient CHT014 (Figure 4D), the EOT PET/CT was ambiguous: a new adenopathy was observed in a low SUV_max_, while LiqBio–MRD was persistently positive. A control PET/CT a month later confirmed progression.

## 4. Discussion

In the current study, by sequencing cfDNA from plasma and gDNA from tissue samples with a targeted panel of 56 genes associated with lymphomagenesis, we characterized the mutational profiles of BCL at diagnosis. Although the number of mutations identified in tissue samples doubled the ones identified in plasma, a correlation for VAF somatic mutations was observed between both samples. Tissue biopsy is still the gold standard technique for lymphoma genetic profiling and diagnosis; however, we have demonstrated that profiling of plasma cfDNA allows the identification of mutations that may not be detectable in tissue biopsies. This could be due to certain clones not being accessible at the biopsy sites because of tumor heterogeneity [16,29]. We identified at diagnosis alterations that were suitable for MRD monitoring, even in plasma cfDNA samples where no tissue biopsies were available, or no mutations were present in paired tissues. Therefore, plasma cfDNA is not a replacement for a tissue biopsy, but it represents a complementary source of tumor DNA for BCL genotyping at diagnosis.

Consistently with previous studies, in FL at diagnosis, we observed the following most recurrently mutated genes (both in tissue and plasma samples): *KMT2D*, *CREBBP*, *BCL2*, and *EZH2* [5,30]. Also, in agreement with previous literature, we found that *CARD11* and *KMT2D* were the most frequently mutated genes in the GC phenotype in DLBCL; and *MYD88*, *PIM1*, and *CD79B* were the most frequently mutated genes in the non-GC phenotype [24,31]. Additionally, we confirmed the higher frequency of *MYD88* mutation in 80% (mainly in the hotspot L265P) of our PCNSL tissue and plasma samples [32]. Finally, consistent with recent reports, we confirmed a molecular signature in PMBCL that differed from the ones of other DLBCLs and was characterized by the presence of *STAT1* and/or *SOCS1* mutations in 65–85% of the cases [21,22].

The applicability of the LiqBio–MRD test, defined by the percentage of patients with somatic mutations suitable to be used as MRD biomarkers, was 71%. Consistently with previous studies, when subdividing by groups, this applicability increased to >80% in DLBCL and PMBCL [29,31,33], and decreased to <50% in PCNSL, suggesting that the blood–brain barrier could prevent ctDNA from entering the circulation [32,34]. In FL, the applicability was 56%, lower than the one recently described [9,20], probably because of the small cohort size. In line with this, the concentration of basal cfDNA in FL samples was half of the one in LBCL samples. Additionally, we found that high cfDNA levels at diagnosis were significantly associated with increased LDH levels and presence of bulky disease, suggesting a correlation with a higher disease burden [20,35]. In the future, we must increase the sensitivity of our NGS panel for cfDNA at diagnosis, as this panel was initially designed for FFPE tissue samples using a small sample size.

In this study, we also assessed the utility of LiqBio–MRD as a dynamic MRD monitoring tool. In comparison with the PET/CT method, LiqBio–MRD demonstrated a lower sensitivity but a higher specificity. Therefore, the combination of both techniques during EOT response assessment and follow-up would provide a more accurate understanding of the patients’ actual condition and favor future remission. We have not been able to predict early relapses because of the small number of samples collected in the follow-up and the difficulty of defining temporal analysis points. Nonetheless, patients in clinical or radiological relapse also showed positivity for LiqBio–MRD.

## 5. Conclusions

Our results demonstrate that genotyping plasma cfDNA is a useful non-invasive complementary tool for the molecular profiling of BCL at diagnosis. Also, it is a suitable method for MRD detection, except for PCNSL, where plasma cfDNA does not contain sufficient ctDNA.

## Figures and Tables

**Figure 1 cancers-15-04022-f001:**
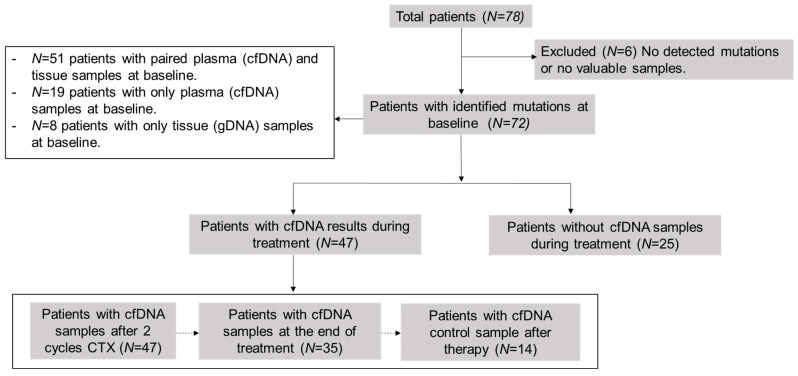
Scheme of analyzed samples. cfDNA = cell-free DNA; CTX = chemotherapy.

**Figure 2 cancers-15-04022-f002:**
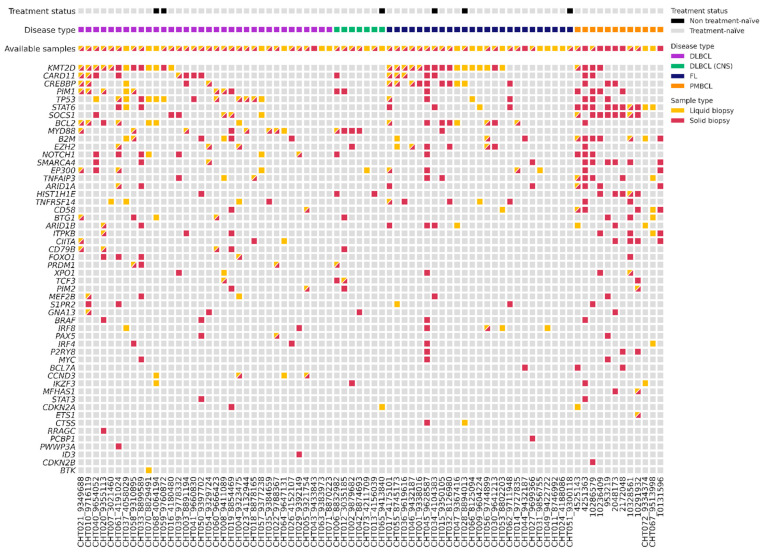
Plot with baseline genotyping of plasma and tissue samples. DLBCL = Diffuse large B-cell lymphoma; DLBCL (CNS) = primary central nervous system lymphoma; FL = follicular lymphoma; PMBCL = primary mediastinal large B-cell lymphoma.

**Figure 3 cancers-15-04022-f003:**
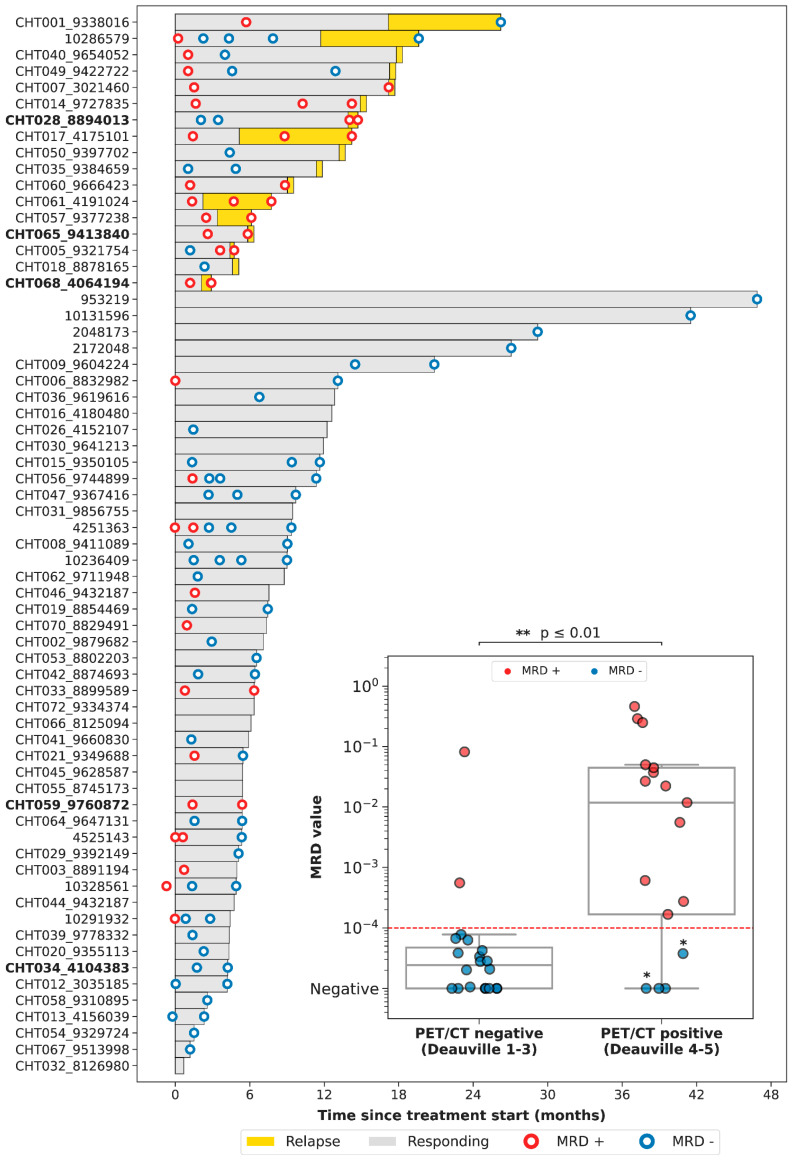
Swimmer plot of the different follow-up time-points. We present a boxplot Internal panel with correlation of LiqBio–MRD and PET/CT. Red dotted lines represent the sensitivity LiqBio–MRD test (2 × 10^−4^). Two data points in blue (*), despite PET positivity, were LiqBio–MRD negative and did not progress. In bold, non-treatment-naïve patients.

**Figure 4 cancers-15-04022-f004:**
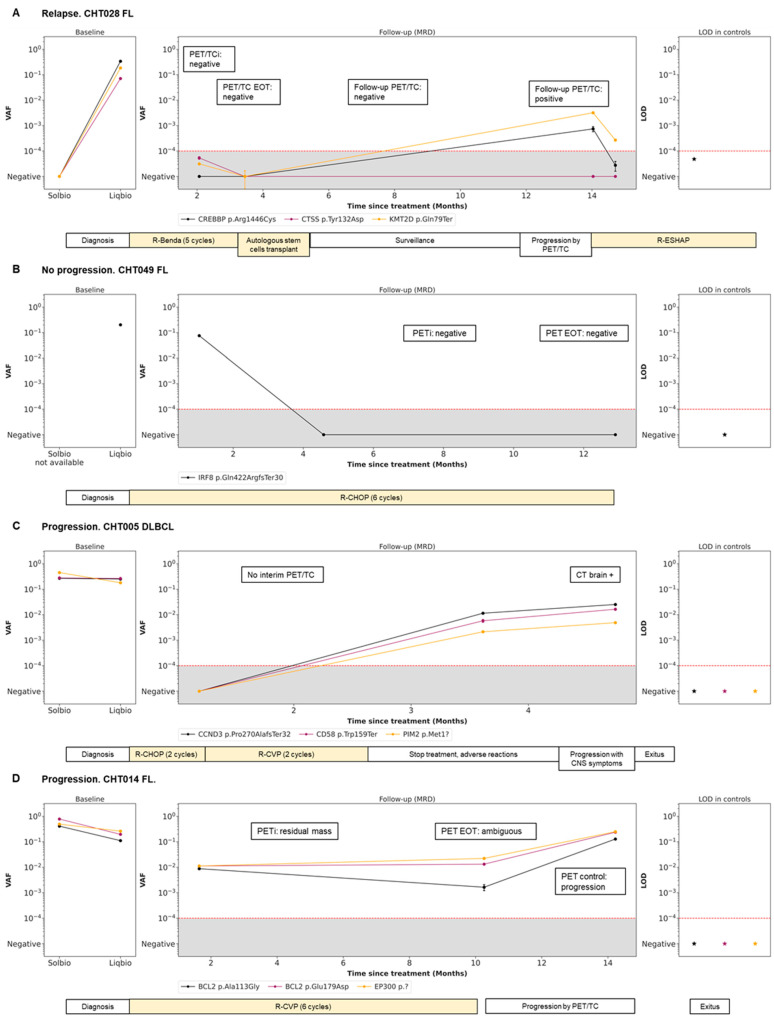
Disease dynamics during follow-up with cfDNA LiqBio–MRD. (**A**–**D**) Baseline genotyping of tissue and plasma cfDNA samples are represented on the left panel (sensitivity 2 × 10^−2^). The central panel represents the variant allele frequency (VAF) values of the different mutations obtained by the ultrasensitive LiqBio–MRD test (sensitivity 2 × 10^−4^). The right panel represents the limit of detection (LOD; mean + 3 SD) defined in healthy control datapoints for every tracked mutation (star). Mutations with LOD above 1 × 10^−4^, represented with red dotted lines, were not used for MRD value calculation.

**Table 1 cancers-15-04022-t001:** Summary of clinical features of B-cell lymphoma cohort (*n* = 78). FL = follicular lymphoma; LBCL = large B-cell lymphoma; PCNSL = primary central nervous system lymphoma. R-CHOP = rituximab/cyclophosphamide/doxorubicin/vincristine/prednisone; DA-EPOCH-R = etoposide/prednisone/vincristine/cyclophosphamide/rituximab; R-Benda = rituximab/bendamustine; R-COP = rituximab/cyclophosphamide/vincristine/prednisone; MTX = methotrexate. ⏀ = Not applicable. FLIPI: Follicular Lymphoma International Prognostic Index. R-IPI: Revised International Prognostic Index. IELSG: International Extranodal Lymphoma Study Group index.

	FL	LBCL	PCNSL	Total Evaluable
Sample size (*n*)	25 (32%)	46 (59%)	7 (9%)	78
Age (median)	62 (41–85)	66 (19–89)	53 (37–77)	62 (19–89)
Sex				
- Male	11 (44%)	22 (48%)	3 (43%)	36 (46%)
- Female	14 (66%)	24 (52%)	4 (47%)	42 (54%)
Lines of treatment				
- Watch and Wait/unfit	1 (4%)	0	0	1 (1%)
- 1st line therapy	21 (84%)	44 (95%)	6 (86%)	71 (91%)
- >1st line therapy	3 (12%)	2 (4%)	1 (14%)	6 (8%)
Transformation	⏀		⏀	
Yes		8 (17%)		
No		36 (78%)		
Ann Arbor stage			⏀	
I–II	6 (24%)	21 (46%)		27 (35%)
III–IV	19 (76%)	23 (50%)		42 (54%)
ECOG				
0–2	24 (96%)	46 (100%)	6 (86%)	76 (97%)
>2	1 (4%)	0	1 (14%)	2 (3%)
Bulky			⏀	
No	17 (68%)	21 (46%)		38 (49%)
Yes	6 (24%)	23 (50%)		29 (37%)
Extranodal involvement			⏀	
Yes	14 (56%)	31 (67%)		45 (58%)
No	11 (44%)	15 (33%)		26 (33%)
Bone marrow involvement				
No	10 (40%)	37 (80%)	7 (100%)	54 (69%)
Yes	14 (56%)	6 (13%)	0	20 (26%)
Hemoglobin (g/dL)				
≤12	5 (20%)	23 (50%)	2 (29%)	30 (38%)
>12	20 (80%)	23 (50%)	5 (71%)	48 (62%)
LDH				
Elevated	10 (40%)	27 (59%)	5 (71%)	42 (54%)
Normal	14 (56%)	19 (41%)	2 (29%)	35 (45%)
β2-microglobulin				
Elevated	14 (56%)	28 (61%)	3 (60%)	45 (58%)
Normal	10 (40%)	16 (35%)	2 (29%)	28 (36%)
FLIPI/R-IPI/IELSG				
Low risk	5 (20%)	17 (37%)	4 (57%)	26 (33%)
Intermediate risk	9 (36%)	24 (52%)	2 (29%)	35 (45%)
High risk	11 (44%)	5 (11%)	1 (14%)	17 (19%)
Chemotherapy				
Rituximab	1 (4%)	0	0	1 (1%)
R-CHOP	10 (40%)	24 (52%)	0	34 (44%)
R-COP	0	1 (2%)	0	1 (1%)
R-Benda	10 (40%)	0	0	10 (13%)
DA-EPOCH-R	0	18 (39%)	0	18 (23%)
MTX-based regime	0	0	7 (100%)	7 (9%)
Others	2 (8%)	2 (4%)	0	4 (5%)
Radiotherapy				
Yes	4 (16%)	14 (30%)	0	18 (23%)
No	21 (84%)	32 (70%)	7 (100%)	60 (77%)
CNS prophylaxis			⏀	
Yes	3 (12%)	18 (39%)		21 (27%)
No	22 (88%)	28 (61%)		50 (64%)
Response				
Complete response	20 (80%)	24 (52%)	5 (71%)	49 (63%)
Partial response	2 (8%)	6 (13%)	0	8 (10%)
Stable disease/progression	2 (8%)	9 (20%)	1 (14%)	12 (15%)

## Data Availability

The data presented in this study are available in this article.

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
