# Peer review of "Genetic Profiling of Cell-Free DNA in Liquid Biopsies: A Complementary Tool for the Diagnosis of B-Cell Lymphomas and the Surveillance of Measurable Residual Disease"

_cancers, 2023, doi:10.3390/cancers15164022_

Round 1

Reviewer 1 Report

In this manuscript, Figaredo et al. compare genetic profiling of plasma cell-free DNA (cfDNA) from liquid biopsies (LiqBio) with standard of care profiling from tissue biobpsie revealing a high correlation between somatic mutations in paired LiqBio and tissue biopsies at diagnosis. They revealed also additional mutations in LiqBio at diagnosis from patients with no available or no mutations detected in tissue samples. They concluded that for MRD detection, LiqBio combined with PET/CT represents a reliable tool to improve PET/CT specificity to identify patients with high risk of relapse.

This manuscript is well written and the study is well designed. It adds important data on the feasibility of LiqBio for patient's management at diagnosis and monitoring.

For these reasons it could be accepted for publication in Cancers at the present form.

Author Response

Dear Reviewer, thank you for your positive feedback.

Reviewer 2 Report

Here are my minor comments on the study:

1. The authors should comment on the downsides of using already existing panels for the study in more detail.

2. Authors should add a comment on the feasibility of use of whole genome sequencing approach in this settings.

3. Would adding copy number signal improve the cfDNA read out and aid the point mutations approach used in the study?

No Comment

Author Response

  1. The authors should comment on the downsides of using already existing panels for the study in more detail.

Dear Reviewer, thank you for your comment. The main reason to use targeted panels is to reduce the costs per patient. We firmly believe a Clinically relevant MRD test needs to be highly applicable and cost-effective. We detect somatic Mutations in 92% of the patients with our targeted approach, and the cost per patient is significantly lower than other High Throughput approaches.

  1. Authors should add a comment on the feasibility of use of whole genome sequencing approach in this settings.

Following the previous answer, the targeted approach is more cost-effective. Others (Kurtz et al. Nature Biotechnology. 2021) used WGS to identify phased mutations. Although these biomarkers permit increasing the MRD test's detection level to 1e-5, analyzing WGS from FFPE samples is challenging. More importantly, from the initial 10mL of PB we obtain 4-5mL of plasma with an average of 17ng/mL. This quantity is not sufficient to aim the level of detection of 1e-5.

  1. Would adding copy number signal improve the cfDNA read out and aid the point mutations approach used in the study?

Excellent question. We explore the possibility of including Copy number alterations (CNVs) as biomarkers. Still, unfortunately, our deep-sequencing LiqBio-MRD test cannot be used with a level of detection of 1e-4 for these biomarkers. We believe the sensitivity limitation for CNAs is common for most NGS-based MRD tests.

Reviewer 3 Report

Thank you for the authors for this interesting paper.

Some clarifications/corrections are still needed.

-In the introduction (BCL) is written twice the whole meaning and th abbreviation. Please remove the latter and use only the selected abbreviation after that in the text.

 - QT is used as an abbreviation for chemotherapy (Figure 1). More often is used chemo/CTX of CTx which are more understandable to all readers. Please change.

-Table 1: Please add the unit for Hemoglobin, and remove the last e from it to be an English term. Regime should be regimen. Intermedian is usually intermediate. Profilaxis is prophylaxis in English. Parcial is partial in English. Non-response would be stable. Is there a reason why PNCSL is separated in this table? The reason is not explained. From R-benda is missing the righ percentage, only (%) is written. From R-CHOP and DA-EPOCH-R and R-COP is missing rituximab from the table text. Please add the explanation for FLIP/R-IPI/IELSG alson in the table text. The term Total would be better as Total of evaluable, as not all are counted in this figure.

-page 5: More informative would be the diagnoses for the six exceptions of treatment-naïve population. Please add. From DA-EPOCH-R and R-COP is missing rituximab. Radiotherapy as monotherapy, not in. Adjuvant radiotherapy is not normally used in lymphoma, the correct term would be consolidative therapy.

-page  6: Why the partial response and progression were grouped? Clinically these are also different diseases.

-Figure 2: DLBCL (SNC) might be CNS-lymphoma? Then the abbreviation would be DLBCL (CNS), as it is in the text but not in the figure itself. Please correct. Please highlight those cases that were not treatment-naïve (then the casenumbers could be taken out of the previous text). Were there any differences between relapsed and treatment-naïve patients?

-Paragraph 3.3. Was the difference found between FL and LBCL or between LBCL subtypes as there is also variation in the aggressiviness of the lymphoma?

-If FL and LBCL were analyzed separately would the correlation results (paragraph 3.3) be different? These are very different cancers.

-Page 7: One relapsed as a Hodgkin's lymphoma. Was this a grey zone case? Please clarify. Or should the term be transformed?

-Page 8: Again in the figure 3, please highlight the non-treatment-naïve patients.

-How the other cancers effect the reliability of this test LiqBio-MRD?

-The sensivity (detection limit) of the test should be written in the "Methods" -section, not only in the figure text 4.

-Please discuss also the "first MRD negative then MRD positive" -scenario. That is the prognostic value of the test?

-Figure 4: the explanations for measured MRD are so small that they are unreadable. Please make those bigger. The diagnosis of the cases should be explained also in here.

-Discussion: the sensitivity results are written here. Usually there should not be any more new results in this paragraph. Please change these results to Results section. How the sensitivity differs from the other tests that are commercially available? Should the combination be used also in the diagnosis phase to ease the later phases, not only in EOT. Why the recommendation to use the test is done to all BCL patients as we know that the FL patients would be treated only when symptomatic in relapse? What would be the clinical value of the information that patient will relapse in the near future if there are no imaging findings/symptoms?

Above.
